# Dietary Supplement Use Among Chinese Primary School Students: A Cross-Sectional Study in Hunan Province

**DOI:** 10.3390/ijerph16030374

**Published:** 2019-01-29

**Authors:** Hanmei Liu, Shiya Zhang, Hanshuang Zou, Yuanlin Pan, Qiping Yang, Yufeng Ouyang, Jing Luo, Qian Lin

**Affiliations:** 1Department of Nutrition Science and Food Hygiene, Xiangya School of Public Health, Central South University, 110 Xiangya Rd, Changsha 410078, China; hanmeiliu@csu.edu.cn (H.L.); yangqiping12@csu.edu.cn (Q.Y.); oyyf0102@csu.edu.cn (Y.O.); luojing2546@csu.edu.cn (J.L.); 2Xiangya School of Public Health, Central South University, 110 Xiangya Rd, Changsha 410078, China; zhangshiya0301@csu.edu.cn (S.Z.); zouhanshuang@csu.edu.cn (H.Z.); l1650490519@csu.edu.cn (Y.P.)

**Keywords:** dietary supplements, primary school, children, China

## Abstract

Dietary supplement use is increasing globally, especially among children. However, few studies have been conducted to examine dietary supplement use among Chinese children. The aim of this study was to examine the prevalence of dietary supplement use and the associated factors in primary school students. A cross-sectional study was conducted in four primary schools in Hunan province, China. A total of 706 Chinese parents/caregivers of school children aged 6–12 years old were enrolled. Self-administrated questionnaires were used to collect information about the children’s dietary supplement use, and their socio-demographic characteristics. The prevalence of dietary supplement use in primary school students was 20.4%. Calcium (16.7%), vitamin C (9.2%), and vitamin D (8.5%) were the most commonly used types of dietary supplement. The main reasons for parents/caregivers to purchase dietary supplements for their children were immunity improvement (43.6%), and for growth promotion (36.5%). Some caregivers (26.4%) purchased dietary supplements online, including from dubious or unknown sellers. However, 37.5% of caregivers did not receive formal guidance on supplementation even if they purchased their supplements through formal channels. Parents/caregivers were unclear about the appropriate choices for children’s dietary supplement use. Guidelines and counseling regarding dietary supplements for children may be helpful in public health work.

## 1. Introduction

There is no specific legal definition of the term “dietary supplement” in China. It generally refers to health products, such as nutrient supplements and health foods, with specific functional claims. The most common health product used by children is nutrient supplements. According to the health product data from 1996–2010 in China, 355 different dietary supplements were able to be used by children, accounting for 85% of children’s health products [1]. As the economy and quality of life have improved, people have begun paying more attention to their health. With the increasing availability and range of health products and nutrient supplements, people are becoming more willing to choose and buy dietary supplements for their families or children.

Dietary supplements represent an important source of essential nutrients. Used correctly, they can decrease nutrient deficiencies and improve health during certain life stage [2]. In the United States, nearly one-third of children and adolescents used dietary supplements (NHANES 2007–2010, 31%) [3], and the use of dietary supplements is also very common for children and adolescents in several Asian countries, such as Korea and Japan [4,5,6]. The prevalence of dietary supplement use among younger age groups is comparatively higher than in older age groups. In China, few studies have thus far discussed the use of dietary supplements in primary school students. Approximately 31.6% to 33.4% of children aged 3–12 years old report using dietary supplements in China [7].

For children, the choice of dietary supplement is often determined by their parents. However, are these products selected by parents all necessary or all reasonable? It is likely that the dietary supplements will achieve the expected goals? The answer is uncertain. Data from the National Health and Nutrition Examination Survey 2003–2006, conducted in America, showed that even with the use of supplements, more than one-third of children failed to meet calcium and vitamin D recommendations among children and adolescents aged 2–18 years [8]. On the other hand, another study showed that multivitamin users were more likely to have potentially excessive intakes, particularly for iron, zinc, vitamin A, and niacin [9]. One study demonstrated that children and adolescents (<19 years old) who used dietary supplements had higher intakes of certain nutrients than non-users [10], which means children who are users may have a lower prevalence of nutrient inadequacy than non-users. It is unclear whether supplements provide an excess of nutrients to children who use dietary supplements. In Changsha city, China, two studies found that a small number of children have high blood calcium level; specifically, 9.8% of children between the ages of 6 to 14 years were estimated to have excess calcium in their blood, suggesting that these children may have had an excessive intake of calcium [11,12].

Most of the current research has focused on dietary supplement interventions for nutritional deficiencies in China. In 2010–2012, a large sample study of Chinese people aged 6–11 years and 60 years and older showed that only 0.71% of participants had used dietary supplements in the previous month [13]. However, there is no data on the use of dietary supplements for healthy children. Therefore, it is difficult to assess the rationality of the use of dietary supplements in children. Caregivers’ perceptions of dietary supplements, along with general access to dietary supplements, remain unclear. There is no specific guideline for the rational use of dietary supplements for Chinese children. Therefore, this study aims to: (1) Assess the prevalence and characteristics of dietary supplement use among healthy Chinese school children; and (2) investigate caregivers’ perceptions regarding dietary supplements for children. This study will provide a scientific basis for the development of guidelines for dietary supplements, alongside relevant policies.

## 2. Materials and Methods

### 2.1. Subjects

Our study used multi-stage stratified cluster sampling. The study took place in Yueyang city and Changsha city, which are located at the western edge of Hunan Province. Four primary schools were selected from these two cities. The sample size was calculated using the PASS software package program (version 11.0 for Windows; NCSS LLC: Kaysville, UT, USA), with an expected prevalence of dietary supplement use of 31.6% [5] and an allowable error of 3.5%. The sample size required was therefore found to be 678. Taking into consideration the potential rates of no response or other non-compliance, 767 participants were recruited, and the final number of valid respondents was 706. The inclusion criteria were good health, age 18 or older, no disease, and being able to read and fill out the questionnaire correctly. The study was approved by the Ethics Review Committee of the Xiangya School of Public Health of Central South University (No. XYGW-2017-41), and all participants signed to indicate informed consent before the investigation.

### 2.2. Variables

The questionnaire consists of 58 items, organized into three parts: (1) Caregivers of primary school students (gender, age, educational level, etc.), basic family situation (average monthly income, relationship to children, the caregivers’ occupations, etc.), and primary school students (age, gender, only-child or not, weight status, child growth standard (based on the 2006 World Health Organization child growth standard as a reference, the age-specific body mass index Z score of BAZ <–2 for weight loss, BAZ >1 for overweight, or BAZ >2 for obesity is calculated)). (2) Dietary supplement use among children: The caregivers were asked whether their child was given any type of supplement for longer than three months during the past year, and to identify the types of dietary supplements taken by their children currently. Dietary supplements include single vitamins, multivitamins, single minerals, multi-minerals, multi-vitamin minerals, and other dietary supplements (including fish oil, amino acids, etc.). (3) Caregivers’ perceptions of dietary supplement use (including supplement information from the provider, procurement channels, the purpose of dietary supplements use, etc.). Daigou is a channel of commerce in which a person outside of China purchases commodities for a customer in mainland China. Daigou commodities are mainly luxury goods, but can also include groceries [14]. In this study, it refers to the purchase of dietary supplements in this way. The questionnaires were distributed and collected by the teachers in each class, and the investigators checked the questionnaire.

### 2.3. Statistical Analysis

EpiData 3.0 software (The EpiData Association, Odense, Denmark) was used for data entry, and the IBM SPSS 18.0 software package (IBM Corp., Armonk, NY, United Sates) was used for statistical descriptions, chi-square analysis tests, and logistic regression analyses. The chi-square test was used to assess the significant differences in the use of dietary supplements depending on demographic data, and the significant differences in parents’ perceptions in different regions. Multiple logistic analyses were employed to examine the independent relationships between dietary supplements, and their 95% confidence intervals were computed from these models. The use rate of different types of dietary supplements is expressed as a percentage of N (%). The test level is set to α = 0.05.

## 3. Results

A total of 706 caregivers participated in this survey. Of these, 70.4% of the participants were women, most caregivers were 31–40 years old (67.9%), BMI was 22.27 ± 3.20, and nearly half of the parents had a college degree or above (47.8%). In our study, most of the caregivers were the children’s parents (94.3%). Only 14.0% of caregivers were involved in a medical occupation. As for the children, 45.4% were boys and 54.6% were girls. The age of the children was 9.06 ± 1.67 years. Nearly half of the primary school students were an only child (42.5%). A total of 7.9% of the children were underweight, and the percentages of overweight and obese children were 14.6% and 11.3%, respectively.

Table 1 shows the use of dietary supplements based on sociodemographic data. Approximately 20.4% of primary school students used dietary supplements. A total of 20.1% of the boys used dietary supplements, and 20.2% of the girls. The percentage of boys and girls using supplements was roughly equivalent (χ^2^ = 0.00, df = 1, α = 0.976). For those children whose caregivers were parents, 21.4% of the children used dietary supplements for more than three months in past year; a significantly higher percentage than the children whose caregivers were not parents (7.4%) (χ^2^ = 3.98, df = 1, α = 0.046). Of the children whose caregivers worked in a medical occupation, 31.0% used dietary supplements, which was significantly higher than those whose caregivers’ occupations were non-medical (18.2%) (χ^2^ = 8.69, df = 1, α = 0.003). Dietary supplement use was different among parents with different educational levels (χ^2^ = 6.65, df = 1, α = 0.036).

As shown in Table 2, compared with caregivers with a college education, caregivers with a middle school education and below, and those with a high school education, were likely to use fewer dietary supplements (AOR = 0.30, 95% CI 0.13–0.71; AOR = 0.54, 95% CI 0.34–0.91). Caregivers working in the medical field were more likely to provide dietary supplements to their children than non-medical workers (AOR = 1.90, 95% CI 1.08–3.33).

Figure 1 shows that the most frequently used supplements were calcium (16.7%), vitamin C (9.2%), vitamin D (8.5%), iron (7.5%), and vitamin E (6.0%). Among the mineral dietary supplements, calcium is the most common supplement, used by 16.7% of children, followed by iron (7.5%). Among the vitamin dietary supplements, vitamin C is the most used (9.1%), and vitamin D is the second most used (8.5%). Nearly one-third of the children taking calcium supplements were also taking vitamin D. Only 1.1% of primary school students used vitamin B supplements. The percentage of use for multi-minerals, multi-vitamins, multi-vitamin minerals, and other dietary supplements was 2.2%, 3.9%, 2.7%, and 2.9%, respectively. There were no significant differences in supplement types used between boys and girls. 

Table 3 shows that 49.1% of caregivers received supplement information from medical staff, followed by television broadcasting and the internet (32.4%), while the direct sale of health products accounted for 7.9%. Most caregivers (71.4%) believe that dietary supplements are necessary for the growth of school-age children, and more than half (66.9%) of caregivers believe that school-age children need to use dietary supplements.

A small number of caregivers believe that dietary supplements can replace a balanced diet (17.1%). A total of 76.5% of caregivers believe that dietary supplements have toxic side effects. Half of the dietary supplements in our study were purchased at pharmacies (52.5%), 21.2% were purchased in hospitals, 11.7% were purchased by Daigou or as online purchases, and 6.4% through direct selling, while 8.3% of these products were gifts. 

Daigou: A channel of commerce in which a person outside of China purchases commodities for a customer in mainland China.

Among the total observed samples, the channels for caregivers to obtain dietary supplement knowledge were firstly from medical staff (49.1%), followed by television, radio and internet (32.4%), a senior or relative (25.9%), professional books (25.6%), professional lectures (14.2%), newspapers, magazines and brochures (9.1%), and direct sales of health products (7.9%) (Table 3).

Among the caregivers who obtained dietary supplements in pharmacies or hospitals (*n* = 240), only 37.4% received instructions about dietary supplement use.

Among caregivers who had purchased dietary supplements for their children, 43.6% reported using supplements to “enhance immunity”, 36.5% to “promote physical growth”, 8.3% to “promote intellectual development”, 5.8% for “supplementing the diet”, and 1.7% because “other kids are using them”.

## 4. Discussion

Dietary supplements play a central role in alternative and complementary medicine [15]. Many varieties of dietary supplements for children are now marketed in China, including single-ingredient products and various combinations of vitamins, minerals, and other constituents. The use of dietary supplements was common among primary school students in our study. There is no doubt that dietary supplements have become widely used in children [16], making it essential for us to consider the safety of these products for children.

Our study found that 20.4% of the observed children (aged 5–12 years old) used dietary supplements, which is lower than reports from other countries, such as the United States (37% of children aged 0–17) [17], South Korea (41% of children between the ages of 1 and 8 [5]), and Japan (40% of children aged 6–12 years old) [6]. The samples from the United States, South Korea, and Japan included both children and teenagers. The results of this study were similar to other studies in China. The results of Wang et al. showed that 16.9–24.2% of children aged 7–12 years used dietary supplements [5]. Another study found that about 22% of primary school students used dietary supplements in Taiwan [18]. Dietary supplement users were defined differently in other studies. In the United States, users were those who had used dietary supplements in the past 30 days. In South Korea, users were those who had used dietary supplements for more than one year in. In Japan, users were those who had used any type of supplement in the past month (present use), or the past year, excluding the past month (past use). Wang recorded the use of dietary supplements over the past six months, and Chen did not explain how users were defined. In our study, dietary supplement use was defined as those who “used dietary supplements for more than three months in the past year”. Considering the occasional use of dietary supplements has little impact on health, only the children who had used dietary supplements for a longer period of time were defined as “dietary supplement users”.

Calcium and vitamin D supplements were popular among children in our study. Calcium and vitamin D are indispensable nutrients for children and adolescents. Calcium is closely related to bone health, and vitamin D can promote the absorption of calcium [19]. Insufficient dietary calcium intake and a low use rate of dairy products may lead to calcium deficiency [20]. The average dietary calcium intake is extremely low in Chinese children and adolescents; the prevalence of calcium intake below the Estimated Average Requirement (EAR) is currently above 96% [21]. In China, calcium deficiency is consequently very common among children—approximately 19.6–34.3% of children have a calcium deficiency [22]. Vitamin D can be produced by the ultraviolet light in sunlight, so the prevalence of vitamin D deficiency varies by region and season, but even in sunny southern China, vitamin D deficiency is common [23]. Approximately 10.8–39.0% of children aged 0–12 years have reported vitamin D deficiency or insufficiency in south China [24,25]. At present, health care providers are more likely to recommend dietary supplements to children under 2-years-old than to other age groups [3]. However, some studies have shown that older children seem to have lower levels of vitamin D serum 25 (OH)D, both at home and abroad [26,27]. This means that the strategy vitamin D supplementation should not be ignored for children over 2 years old. Appropriate guidance could be provided to improve the nutritional status of children with calcium or vitamin D deficiency, such as advising an increase in the intake of milk or other calcium-containing foods, increasing children’s outdoor activities, and so on. In combination with such advice, dietary supplements can be given to supplement inadequate dietary intake if necessary [28]. Dietary quality has been greatly improved in recent years among Chinese residents, however in addition to calcium deficiency, some studies have found hypercalcemia in children. At the same location of this survey, a study found that only 1.2% of children aged 6–14 had a calcium deficiency, while 9.8% had a high blood calcium level [11]. The author thought that this result was probably caused by the overuse of calcium supplements. There is still controversy about whether calcium and vitamin D use can prevent disease, and whether there can be adverse reactions to supplementation in healthy people [29,30]. Therefore, individual nutritional levels should be taken into account for calcium and vitamin D supplementation, to avoid the health problems of overuse.

The status of iron deficiency and the rate of anemia have also improved recently. Although much attention has been paid to iron deficiency and anemia, it can still be a serious issue, especially in rural areas [31,32]. Approximately 35.5% of children aged 3–12 years in seven Chinese cities and two countryside areas were found to have an iron deficiency [33]. Iron deficiency can lead to anemia, physical retardation, and even limit the development of children’s intellectual abilities [34]. Moreover, inadequate iron intake can also limit the absorption of some important nutrients, such as zinc [35]. Therefore, in order to improve iron deficiency and anemia, we should pay more attention to children with an iron deficiency and those who live in rural areas. Furthermore, vitamin C can promote iron absorption [36]. The widespread use of vitamin C might be because of the high rate of the iron-deficiency anemia. With the change of dietary structure of Chinese residents, the proportion of fruits and vegetables in the diet has decreased over time. Dietary vitamin C intake has decreased among Chinese teenagers, especially in low-income rural families [21,37]. Vitamin C is closely related to dietary structure, and it is necessary to strengthen nutrition education and intervention for children and their caregivers. Strategies could include correcting unreasonable dietary habits among children who are picky and partial eaters, adjusting the dietary structure of children, and encouraging the increased intake of vegetables and fruits. Vitamin B is also important for school-age children; vitamin B12 levels are related to neurocognitive functions in school-age children [38]. Unfortunately, few studies have investigated the prevalence of vitamin B deficiency in healthy children [22]. One study showed that vitamin B intake tends to be inadequate from food alone [39]. Our survey showed that the percentage of vitamin B use was only 1.1%. We need to pay more attention to vitamin B deficiency among Chinese children. School-age children are in a critical period of growth and development, and for those populations who have nutrient deficiencies, supplementing nutrients appropriately could improve physical growth, nutritional status, and cognitive ability, especially among school-age children who cannot intake adequate nutrients from food [28,40,41].

There was no significant difference in dietary supplement use between different age groups in our study, a finding that differs from those of other studies [4]. It may be that the participants in our study were all primary school students aged 4–12 years, excluding infants and adolescents. In our study, several demographic characteristics of caregivers were related to supplement use. Caregivers who had college degrees or above, and those in a medical occupation, tended to use dietary supplements more frequently. Caregivers with higher education and working in a medical occupation may have greater health awareness. Some studies have suggested that parents’ health behavior may affect their children, and parents who use dietary supplements themselves are more likely to give their children dietary supplements [17]. Studies have also shown that dietary supplement use is associated with a healthy lifestyle. People who use supplements are more likely to engage in physical activities or have better dietary habits than those who do not use them [42,43]. There have also been studies that suggest that users of supplements obtain a more comprehensive amount of nutrients from food than non-users [44], which means that the nutrients obtained from food are probably already sufficient for children who use supplements [45]. At present, there is little data to support the idea that dietary supplement use can promote health for those who already intake sufficient nutrients, and regarding whether dietary supplements in this group simply provide an excess of nutrients. This may be revealed by future investigations.

As we outlined earlier, approximately one-third of caregivers believe that dietary supplements are necessary for primary school students, and most also believe that dietary supplements have toxic side effects. In China, the primary purpose of dietary supplement use is to enhance children’s immunity and promote their physical growth, rather than to supplement the diet. Perhaps caregivers believe it is necessary for their children to take dietary supplements, but do not know which nutrients their children really need. In the United States, the primary reason for dietary supplement use was to improve or maintain health, rather than enhance immunity. Conversely, only a small number of children were given dietary supplements to enhance nutrient intake [3]. This means most caregivers have inadequate knowledge regarding the function of dietary supplements. Only half of the supplement use reported was based on recommendations from medical staff (49.1%). This suggests that there are many caregivers who do not receive formal instructions for dietary supplement use. In this survey, caregivers preferred to purchase dietary supplements through hospitals and pharmacies. However, more than half of the surveyed parents did not learn about dietary supplements from a formal supplement provider (such as medical staff, professional books, lectures, etc.), even if they bought them at the pharmacy. In addition, parents with college degrees are more likely to use dietary supplements. However, among these caregivers, 32.3% purchased dietary supplements through online shopping, Daigou, health product direct selling, or other informal channels. Online shoppers tend to focus on the shopping experience (such as price) when choosing products, and might ignore whether the products are truly suitable [46], potentially leading to poor choices in dietary supplements. In China, Daigou health care products have occupied a large market. Some caregivers lack the correct guidance and blindly trust Daigou products, however the sources and ingredients of these products are unclear. Furthermore, these products have not been inspected by the national customs and quality inspection department. There are real questions regarding the safety of these products, particularly for those using dietary supplements regularly.

There are some limitations to this study. Firstly, the population in our study cannot reflect the dietary supplement use of all Chinese children. Secondly, we did not investigate dietary intake to assess whether the overall nutrient intake among the children in our study was appropriate. However, the results of this study can provide a rough picture of dietary supplement use among school-age children in China, and help to improve caregivers’ perceptions and purchase behaviors around dietary supplements. More importantly, it may provide a reference for the monitoring and guidance of dietary supplements for Chinese children.

## 5. Conclusions

In this study, we found that 20.4% of primary school students used dietary supplements. The educational background of caregivers and the occupation of caregivers were both related to the use of dietary supplements for children. The most popular dietary supplements are calcium, vitamin C, vitamin D, iron, and vitamin E. Dietary supplements can be used when children are not getting adequate calcium, vitamin C, vitamin D, iron and vitamin E from their diet, and appropriate guidance should be provided for caregivers or children regarding the effective use of supplements.

## Figures and Tables

**Figure 1 ijerph-16-00374-f001:**
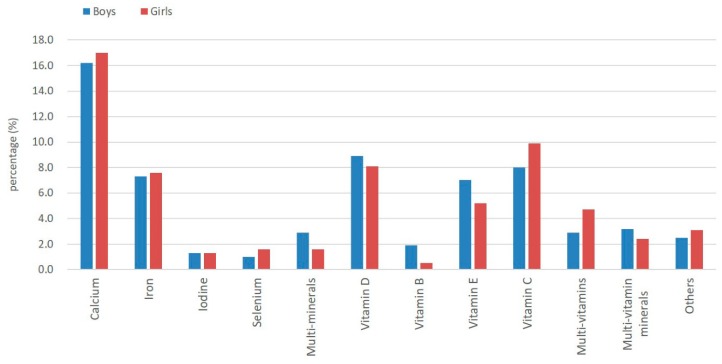
Percentages of different types of dietary supplements among primary students.

**Table 1 ijerph-16-00374-t001:** Dietary supplement use among primary school students by demographic characteristics.

Variables		*N*	Dietary Supplement Use ^#^		χ^2^	*p*-Value
			*n*	(%)		
Total		706	144	20.4		
**Area**	Changsha City	381	84	22.0	1.389	0.239
	Yueyang City	325	60	18.5		
**Caregivers**						
Gender	Males	209	47	22.5	0.800	0.371
	Females	497	97	19.5		
Age (y)	≤30	55	12	21.8	0.805	0.669
	31–40	479	101	21.1		
	>40	172	31	18.0		
Weight status *	Underweight	50	12	24.0	1.862	0.394
	Healthy weight	514	98	19.1		
	Overweight/obese	117	28	23.9		
Educational level *	Junior middle school and below	108	15	13.9	6.649	0.036
	Senior middle school	250	47	18.8		
	College and above	329	81	24.6		
Relationship to children *	Parents	622	133	21.4	3.982	0.046
	others	38	3	7.9		
Occupation *	Medical	86	29	33.7	8.687	0.003
	Non-medical	530	104	19.6		
Average monthly income *	Low income	119	23	19.3	0.386	0.825
	medium income	265	54	20.4		
	High income	134	30	22.4		
**Children**						
Gender *	Boys	314	63	20.1	0.001	0.976
	Girls	382	77	20.2		
Age (y) *	5–6	96	23	24.0	0.960	0.619
	7–9	336	67	19.9		
	10–12	238	46	19.3		
Only-child *	Yes	293	53	18.1	2.057	0.152
	No	399	90	22.6		
Weight status *	Underweight	50	13	26.0	3.702	0.295
	Healthy weight	416	84	20.2		
	Overweight	92	19	20.7		
	Obese	72	9	12.5		

* Missing data not included. ^#^ Dietary supplements use: Used dietary supplements more than three months in past year.

**Table 2 ijerph-16-00374-t002:** Logistic regression analysis of the factors in dietary supplement use among school children.

Variables	Crude OR (95%CI)	Adjusted OR (95%CI)
Area (vs. Changsha)	0.801 (0.553,1.159)	1.161 (0.702,1.921)
Gender of the children (vs. boys)	1.006 (0.693,1.460)	1.022 (0.650,1.605)
Age of the children(years) (vs. 5–6)		
7–9	0.791 (0.461,1.356)	1.355 (0.687,2.673)
10–12	0.760 (0.431,1.343)	1.286 (0.619,2.670)
Only-child (vs. No)	0.758 (0.519,1.108)	0.634 (0.398,1.008)
Children’s Weight status (vs. healthy weight)		
Underweight	1.389 (0.707,2.729)	1.837 (0.842,4.006)
Overweight	1.029 (0.588,1.799)	1.132 (0.604,2.120)
Obese	0.565 (0.270,1.181)	0.660 (0.278,1.567)
Caregivers’ relationship to children (parents vs. others)	3.173 (0.961,10.478)	3.254 (0.740,14.315)
Caregivers’ educational level (vs. college and above) ^a^		
Junior middle school and below	0.494 * (0.271,0.900)	0.299 * (0.126,0.710)
Senior middle school	0.502 (0.473,1.062)	0.535 * (0.315,0.907)
Caregivers’ Occupation (vs. non-medical) ^b^	2.084 * (1.269,3.422)	1.893 * (1.076,3.333)

* *p* < 0.05. ^a^ Adjusted for area, gender of the children, age of the children, only-child, weight status, relationship to children, and occupation. ^b^ Adjusted for area, gender of the children, age of the children, only-child, weight status, relationship to children, and educational level.

**Table 3 ijerph-16-00374-t003:** Caregivers’ perception and purchase channels of dietary supplements for their children.

Variables	Area		Total (*n* = 706)	*P*-Value
	Changsha (*n* = 381)	Yueyang (*n* = 325)		
**Perception to dietary supplements**				
Dietary supplements are necessary for the growth of school-age children ^#^	187(70.0)	168(73.0)	355(71.4)	0.460
School-age children need to use dietary supplements ^#^	175(65.8)	153(68.3)	328(66.9)	0.556
Dietary supplements can replace a balanced diet ^#^	44(14.5)	47(20.4)	91(17.1)	0.072
Dietary supplements have toxic side effects ^#^	158(78.2)	119(74.4)	277(76.5)	0.392
**Purchase channels ^#^**				
Pharmacies	81(47.4)	90(58.1)	171(52.5)	0.022
Hospitals	35(20.5)	34(21.9)	69(21.2)	
Overseas Daigou or online purchases	29(17.0)	9(5.8)	38(11.7)	
Health Products Direct Selling	13(7.6)	8(5.2)	21(6.4)	
Other people’s gifts	13(7.6)	14(9.0)	27(8.3)	
**Dietary supplements knowledge comes from ^#^**				
Medical staff ^#^	185(48.6)	155(49.8)	340(49.1)	0.737
Television broadcasting and the internet ^#^	147(38.6)	77(24.8)	224(32.4)	0.000
Elders or relatives ^#^	104(27.3)	75(24.1)	179(25.9)	0.342
Professional books ^#^	104(27.3)	73(23.5)	177(25.6)	0.251
Professional lectures ^#^	57(15.0)	41(13.2)	98(14.2)	0.505
Newspaper, magazine or bulletin boards ^#^	42(11.0)	21(6.8)	63(9.1)	0.052
Health products direct selling lectures ^#^	39(10.2)	16(5.1)	55(7.9)	0.014

^#^ All categories do not have the same sample sizes due to missing data.

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
