# Peer review of "Dietary Supplement Use Among Chinese Primary School Students: A Cross-Sectional Study in Hunan Province"

_ijerph, 2019, doi:10.3390/ijerph16030374_

Round 1
Reviewer 1 Report
-What is the sample size calculus? Please add it.
-What are variables the odds ratio was adjusted?
-The authors cited the 21 study, however, it is published in low quality journal.
-What is calcium intake among children?
-The authors confirmed in text that appropriate guidance of dietary supplements should be provided for caregivers and children. However, this information is speculative. The authors must describe in discussion and conclusion that dietary supplements may be used if habitual diet is poor in calcium, vitamin c, d, and iron.
Author Response
Dear reviewer,
We would like to thank you for those insightful comments and suggestions. The comments and suggestions have been extremely valuable and helpful for revising and improving our paper. Here we submit a new version of our manuscript with the title “Dietary supplement use among Chinese primary school students: a cross-sectional study in Hunan Province,” (ID: 429737), which has been revised according to the suggestions. Efforts were made to correct the mistakes and improve the English of the manuscript. We also have checked our database, repeated all the analysis, and revised mistakes. You will clearly see the difference made to the revised manuscript.
If you have any question about this paper, please do not hesitate to contact me.
Sincerely yours,
Corresponding author
Qian LIN , Ph.D, Professor E-mail: linqian@csu.edu.cn
Below we detail the point-to-point response to the reviewer’s comments.
Reviewer 1
Point 1: What is the sample size calculus? Please add it.
Response 1: Thank you for pointing out this. It has been documented that the prevalence of dietary supplements is 31.6 among boys and 33.2% among girls (reference 5). We have added the calculation of sample size into the paper, please see page 2, line 79-83.
Corrections: “Sample size was calculated using the PASS software package program (version 11.0 for Windows; NCSS LLC: Kaysville, UT, USA) with an expected prevalence of dietary supplement use of 31.6% [5] and an allowable error of 3.5%. The sample size required was 678. Considering no response or other non-conforming situations, 767 participants were recruited, and the final valid respondents were 706.”
Point 2: What are variables the odds ratio was adjusted?
Response 2: We really appreciate for this comment. We re-analyzed the data we found there were some mistakes in our previous analysis process. We re-wrote the results and modified Table 2 (Page 4, line 135-146). Considered only few caregivers received elementary school education or below, we combined them into the group “Junior middle school and below”.
Corrections: “As Table 2 showed, compared with caregivers with college education, caregivers with middle school education and below, caregiver with senior middle school education were more likely to use fewer dietary supplements (AOR= 0.30, 95% CI 0.13–0.71; AOR= 0.54, 95% CI 0.34–0.91). Compared with non-medical workers, caregivers worked in medical field were more likely to give dietary supplements to their children (AOR= 1.90, 95% CI 1.08–3.33).”
Point 3: The authors cited the 21 study, however, it is published in low quality journal.
Response 3: Thank you for pointing out this. We chose another article published in 2017 to replace the reference. Since the results of different studies on dietary calcium intake are very consistent, calcium deficiency is very common in China, so the discussion on calcium supplements has not changed (Page 8, line 213-215).
Corrections: The average dietary calcium intake was extremely low in the Chinese children and adolescents, the prevalence of calcium intake below the Estimated Average Requirements (EAR) was above 96% [21].
Reference 21--Wang, H.; Wang, D.; Ouyang, Y.; Huang, F.; Ding, G.; Zhang, B., Do Chinese Children Get Enough Micronutrients? Nutrients 2017, 9, (4), 397.
Point 4: What is calcium intake among children?
Response 4: The average dietary calcium intake was extremely low in the Chinese children and adolescents, the prevalence of calcium intake below the Estimated Average Requirements (EAR) was above 96%. (Page 8, line 215). In our study, we did not investigate dietary intake to assess whether calcium and other nutrient intakes are appropriate. We have added two sentences in study limitation, page 10, line 301-305.
Point 5: The authors confirmed in text that appropriate guidance of dietary supplements should be provided for caregivers and children. However, this information is speculative. The authors must describe in discussion and conclusion that dietary supplements may be used if habitual diet is poor in calcium, vitamin c, d, and iron.
Response 5: Very appreciate for these Suggestions, we re-wrote it both in the discussion and conclusions. (Page 8, Line 210-259/ Page 10, Line 312-314).
Corrections: “The calcium and vitamin D were popular among children in our study. Calcium and vitamin D were indispensable nutrients for children and adolescents……that means vitamin D supplementation may cannot be ignored for those children more than 2 years old. Appropriate guidence could be provided to improve their nutritional status for these children with calcium or vitamin D deficiency, such as increasing the intake of milk or other calcium-containing foods, increasing children's outdoor activities and so on. Moreover, dietary supplements can be given to supplement inadequate dietary intake if it is necessary [28]”
“The status of iron deficiency and the rate of anemia were improved recently…. Therefore, in order to improve their iron deficiency and anemia, we could pay more attention to children with iron deficiency and lived in rural areas. Vitamin C can promote iron absorption [36]……We need pay more attention to vitamins B deficiency among Chinese children. School-age children are in a critical period of growth and development, as for those populaiton who have nutrients deficiency, supplementing nutrients appropriately could improve the physical growth, nutritional status or cognitive ability among those school-age children who cannot intake adequate nutrients from food [28,40-41].”
“Dietary supplements can be used when children are not getting adequate calcium, vitamin C, vitamin D, iron and vitamin E from their diet,and appropriate guidance shoud be provide for caregivers or children simultaneously.”

Reviewer 2 Report
I appreciate the opportunity to review this paper. I would like to contribute with the following comments to improve its quality, so the readership may be better informed about the significance of findings.
In the Introduction I found it difficult to differentiate when the authors were referring to studies carried out in China or in other places of the world. Without the context, it is difficult for the reader to scope the actual problem the authors are addressing in this section. Also, it is not clear which of the many caveats in current knowledge will the article address, as even in the last paragraph of the Introduction there are several questions raised, which the article does not attempt to clarify (for example, “it is hard to know the safety and efficacy of dietary supplements use from previous studies”, followed by a sentence starting with “Therefore, this study aims to ….” But then the authors do not address the specific quote.
Presentation of results in text and tables seems not to correspond. For example, the authors mention (Page 3, Line 110-11) that “94.3% of the caregivers were children’s parents”, but Table 1 (page 4shows that 21.4% of “Relationship to children” were Parents. (The end note, which points out that “Missing data are not included”, is not helpful, given the previous figure mentioned in the text.)
Page 6, line 153, seems like a typo: “nearly three/thirds of caregivers’ supplements knowledge …”, corresponding (line 154) to “32.4%”.
Page 6, lines 157-159, need to be better explained, as the figures explaining where the knowledge about dietary supplements comes from seem to be at odds with what is mentioned on previous lines 154-155.
I would like to caution the authors about misleading the public because of lack of precision in their description/comments. For example, it is clear that the study only addresses primary school students in Hunan (not in all of China), but even then the reader is not sure how representative of the province are the two locations with four primary schools selected for the study. This is really critical in the Discussion, when comparisons are made between these findings and those from other studies. Some sentences are not very clear about the representability of the sample, like (Page 7, line 178): “Our research focused on the dietary supplements use in primary school students in China.” Similarly, it is difficult to understand whether other studies refer to national/regional/local representative samples; for example, when referring to References 15, 16, 17. Likewise, it is not always clear what are the ages of the individuals included in different studies. Without further clarification of the methods used, we are not sure if we are comparing “apples to apples”.
Following the description offered, it is clear that the study collected information from the caregivers; however, the language used in some places gives the impression that it is the children who answered. Take for example page 7, line 166: “Among children who took dietary supplements, 43.6% reported using supplements to “enhance …”
The article will benefit from a paragraph in the Discussion where the authors identify and recognize the caveats of their study, including aspects already raised about the representability of the sample, potential biases, the influence of the definition of “users” (and an explanation of why a “relatively strict” definition (Line 191) was decided upon.
Author Response
Dear reviewer,
We would like to thank you for those insightful comments and suggestions. The comments and suggestions have been extremely valuable and helpful for revising and improving our paper. Here we submit a new version of our manuscript with the title “Dietary supplement use among Chinese primary school students: a cross-sectional study in Hunan Province,” (ID: 429737), which has been revised according to the suggestions. Efforts were made to correct the mistakes and improve the English of the manuscript. We also have checked our database, repeated all the analysis, and revised mistakes. You will clearly see the difference made to the revised manuscript.
If you have any question about this paper, please do not hesitate to contact me.
Sincerely yours,
Corresponding author
Qian LIN , Ph.D, Professor E-mail: linqian@csu.edu.cn
Below we detail the point-to-point response to the reviewer’s comments.
Reviewer 2
Point 1: In the Introduction I found it difficult to differentiate when the authors were referring to studies carried out in China or in other places of the world. Without the context, it is difficult for the reader to scope the actual problem the authors are addressing in this section. Also, it is not clear which of the many caveats in current knowledge will the article address, as even in the last paragraph of the Introduction there are several questions raised, which the article does not attempt to clarify (for example, “it is hard to know the safety and efficacy of dietary supplements use from previous studies”, followed by a sentence starting with “Therefore, this study aims to ….” But then the authors do not address the specific quote.
Response 1: Thank you for your suggestion. We revised the article.
Page 1,line 37, we added "in China".
Page 2,line 46, we added " such as the Korea and Japan ".
Page 2,line 48-49, we changed it to " Approximately 31.6% to 33.4% of children aged 3-12 years old reported using dietary supplements in China”
Page 2,line 52-53, we added "A data from National Health and Nutrition Examination Survey 2003-2006 of America".
Page 2,line 61-63, we added detailed regional and data " In Changsha city of China, two study also found that a small number of children have high blood calcium level, for example, 9.8% of children have excess calcium in their whole blood between the ages of 6 to 14 years".
Page 2,line 65-70, we revised the paragraph.
Corrections: " In 2010-2012, a large sample study of Chinese people aged 6-11 and 60 years and older showed that only 0.71% of participants used dietary supplements previous month. [11]. However, there is no data on the use of dietary supplements for healthy children. Therefore, it is difficult to assess the rationality of the use of dietary supplements in children. The caregiver’s perception of dietary supplements and the access to dietary supplements remain unclear. There is no specific guideline for the rational use of dietary supplements for Chinese children."
Point 2: Presentation of results in text and tables seems not to correspond. For example, the authors mention (Page 3, Line 110-111) that “94.3% of the caregivers were children’s parents”, but Table 1 (page 4 shows that 21.4% of “Relationship to children” were Parents. (The end note, which points out that “Missing data are not included”, is not helpful, given the previous figure mentioned in the text).
Response 2: Thank you for your suggestion. Page 3, Line 117-118, the result 94.3% which mentioned in the article refers to the population composition of the parents of the children, while the 21.4% presented in table 1(Page 4)refers to the use rate of dietary supplements among the subjects were parents of their children. We revised the sentence to “For those children whose caregivers were parents, 21.4% of the children used dietary supplements more than 3 months in past year, significantly higher than the children whose caregivers were not parents (7.4%)”. (Page 3, Line 124-126)
Point 3: Page 6, line 153, seems like a typo: “nearly three/thirds of caregivers’ supplements knowledge …”, corresponding (line 154) to “32.4%”.
Response 3: Thank you for your suggestion. We have revised this paragraph, “nearly one and some words changed accordingly.”(Page 6, line 167-170).
Corrections: Among the total observed samples, the channels for caregivers to obtain dietary supplement knowledge were mainly medical staff (49.1%), followed by television, radio and internet (32.4%), and senior or relative (25.9%), professional books (25.6%), professional lectures (14.2%), newspapers, magazines and brochures (9.1%) and health products direct selling (7.9%). (Table3)
Point 4: Page 6, lines 157-159, need to be better explained, as the figures explaining where the knowledge about dietary supplements comes from seem to be at odds with what is mentioned on previous lines 154-155.
Response 5: Thank you for your suggestion. We have revised this paragraph, Page 6-7, line 175-176, and added figure 2 to explain our findings.
Corrections: Among the caregivers who obtained dietary supplements in pharmacies or hospitals (n=240), only 37.4% received instruction about dietary supplements use (Figure 2.).
Point 5: I would like to caution the authors about misleading the public because of lack of precision in their description/comments. For example, it is clear that the study only addresses primary school students in Hunan (not in all of China), but even then the reader is not sure how representative of the province are the two locations with four primary schools selected for the study. This is really critical in the Discussion, when comparisons are made between these findings and those from other studies. Some sentences are not very clear about the representability of the sample, like (Page 7, line 178): “Our research focused on the dietary supplements use in primary school students in China.” Similarly, it is difficult to understand whether other studies refer to national/regional/local representative samples; for example, when referring to References 15, 16, 17. Likewise, it is not always clear what are the ages of the individuals included in different studies. Without further clarification of the methods used, we are not sure if we are comparing “apples to apples”.
Response 5: Thank you for your reminding. Your Suggestions have helped me a lot. I have modified this part according to your suggestion,We rewrote these sentences (Page 8, line 193-196).
In page 1,line 16-18, We have deleted the word “in China” .
In page 7-8,line 199-202,The previous references 15, 16, and 17 have been replaced by references 17, 5, and 6 .
Corrections: Our study found that 20.4% of observed children (aged 5-12 years old) used dietary supplements, which is lower than reports in other countries, such as the United States (37%, children aged 0-17) [17], South Korea (41%), children between the ages of 1 and 8 [5] and Japan (40%, 6) -12 years old children) [6]. (Page 8, line 193-196).
Point 6: Following the description offered, it is clear that the study collected information from the caregivers; however, the language used in some places gives the impression that it is the children who answered. Take for example page 7, line 166: “Among children who took dietary supplements, 43.6% reported using supplements to “enhance …”
Response 6: Thank you for your suggestion. We have modified this part and changed it to " Among caregivers who have purchased dietary supplements for their children, "(Page 7, line 180).
Point 7: The article will benefit from a paragraph in the Discussion where the authors identify and recognize the caveats of their study, including aspects already raised about the representability of the sample, potential biases, the influence of the definition of “users” (and an explanation of why a “relatively strict” definition (Line 191) was decided upon.
Response: Thank you for your suggestions. We admit that the words in this section was not rigorous enough. We have made a revision.
Corrections: In our study, dietary supplements use was defined as “sed dietary supplements for more than 3 months in the past year”. Considering the occasional use of dietary supplements has little impact on health, only the children who have used dietary supplements for a longer time were defined as “dietary supplements users”. (Page 8, line 205-209).

Round 2
Reviewer 1 Report
OK
Author Response
Thank you very much. We have submitted the manuscript to MDPI for language editing.
Reviewer 2 Report
I thank the authors for careful consideration to this Reviewer's previous comments. I understand that English is not the main language of the writing author, so I recommend the article is sent to a professional English-speaking editor, for clarity of concept and ease of understanding.
Author Response
Thank you very much for your suggestions. We have submitted the manuscript to MDPI for language editing. We carefully checked the meaning of the sentences to make sure it was consistent with the original intention.